# The Clinical and Psychopathological Profile of Inpatients with Eating Disorders: Comparing Vomiting, Laxative Abuse, and Combined Purging Behaviors

**DOI:** 10.3390/healthcare12181858

**Published:** 2024-09-15

**Authors:** Matteo Panero, Francesco Bevione, Ilaria Sottosanti, Paola Longo, Federica Toppino, Carlotta De Bacco, Giovanni Abbate-Daga, Matteo Martini

**Affiliations:** Eating Disorders Unit, Department of Neuroscience “Rita Levi Montalcini”, University of Turin, Via Cherasco 15, 10126 Turin, Italy; francesco.bevione@unito.it (F.B.); ilaria.sottosanti@edu.unito.it (I.S.); paola.longo@unito.it (P.L.); federica.toppino@unito.it (F.T.); carlideb@yahoo.it (C.D.B.); giovanni.abbatedaga@unito.it (G.A.-D.); matteo.martini@unito.it (M.M.)

**Keywords:** anorexia nervosa, eating disorders, purging behavior, hospitalization

## Abstract

Background/Objectives: The previous literature on purging behavior in eating disorders (EDs) suggests an overall more complicated clinical picture for individuals with this symptomatology. So far, no studies have analyzed the possible differences between the specific types of purging among ED inpatients. Methods: A clinical sample of 302 inpatients with EDs was classified according to no purging behaviors, vomiting, the abuse of laxatives, and both vomiting and the abuse of laxatives. Participants completed the following questionnaires: the Eating Disorder Examination Questionnaire (EDE-Q), Frost Multidimensional Perfectionism Scale (F-MPS), State–Trait Anxiety Inventory (STAI), and Beck Depression Inventory (BDI). Clinical information was collected for each individual. Results: Significant differences in the four groups were evidenced in age (*p* < 0.001), years of illness (*p* < 0.001), BMI at discharge (*p* < 0.001), STAI state anxiety (*p* < 0.001), STAI trait anxiety (*p* < 0.001), BDI (*p* < 0.001), EDE-Q eating concerns (*p* < 0.001), EDE-Q shape concerns (*p* < 0.001), EDE-Q weight concerns (*p* < 0.001), EDE-Q global score (*p* < 0.001), and F-MPS parental criticism (*p* < 0.001). ED inpatients with purging behaviors were older, had a longer duration of illness, higher parental criticism, and worse general and eating psychopathology. No differences emerged between the specific types of purging behavior. Conclusions: Purging behavior is a marker of severity in EDs independently of the specific type of purging. The appearance of any purging behavior must be regarded as a considerable red flag and be followed by an intensification of the cure.

## 1. Introduction

Eating disorders (EDs) are severe mental disorders causing a heavy impairment of quality of life, whose course is chronic or relapsing in about 25% of affected subjects [1].

Among EDs, anorexia nervosa binge eating/purging type (AN-BP), bulimia nervosa (BN), and purging disorder (PD) are characterized by the presence of purging behaviors aimed at losing weight or preventing weight gain [2]. The main purging behaviors are auto-induced vomiting, laxative abuse, and diuretic abuse. A general transition from restricting to purging pictures of EDs, and a less common transition in reverse, have been identified [3,4]. Also, different diagnoses of EDs seem to be maintained by similar psychopathological processes [5]. This has led some authors to conceive them as different grades across a spectrum rather than separate diagnoses [4,5,6,7]. At the same time, PD showed distinct clinical features from BN and AN [8,9,10,11,12], and ED diagnoses showed different outcomes in the long term [13]. For these reasons, agreement has not yet been reached on the concept of EDs as movable diagnoses across a unique spectrum [14,15].

### 1.1. Purging Behaviors

Auto-induced vomiting consists of the voluntary expulsion of vomit in order not to adsorb food. This produces dehydration and the loss of electrolytes, especially potassium, and leads to mouth and esophageal mucosa being exposed to the stomach content. Hypotension, electrolyte imbalances, and dental damage are the most frequent complications of vomiting [16]. In the most severe cases, parotid sialadenitis, oral bleeding, palate ulcers, facial petechiae, esophagitis, esophageal motility alterations and sudden death are possible consequences [17,18,19,20,21,22]. Laxative abuse is the assumption of unprescribed doses of laxatives to expulse intestinal content. The most common categories of laxatives utilized are stimulants, bulk-forming, saline and hyperosmotic agents, emollients, and lubricants. Different complications may appear based on the drug and the entity of the abuse. The most frequent are diarrhea, abdominal cramps and pain, but volume depletion and electrolyte alterations can occur if the abuse is heavy. Besides laxatives, diuretics are often assumed to help individuals lose liquids and consequently lower weight.

Purging behavior has both physical and psychological effects on individuals. A recent study found that disordered eating behaviors might fulfill four functions: alleviating shape, weight, and eating concerns, regulating emotions, regulating one’s self-concept, and regulating interpersonal relationships/communicating with others [23]. Also, purging behavior was described as a means to relieve gastrointestinal distress or the distress that might follow eating [23]. Another qualitative investigation revealed purging behaviors as attempts to cope and control, improve self-regard and social status, regulate emotion, and provide physiological reinforcement [24]. A decrease in negative affect following purging has been evidenced both in BN and AN [25,26,27,28]. Studies on animals have found a decrease in acetylcholine levels after purging, suggesting this might soothe negative feelings [29]. Different conditions elicit purging behaviors in EDs. In many cases, they are a consequence of binge eating, overeating, or loss of control episodes. Overall, binge eating is the strongest predictor of purging behavior. In particular, in BN, they are linked to the loss of control, while in AN, they are frequently preceded by an increase in negative feelings [29]. Even though it is not possible to assess a causal relationship between them, purging behaviors have been associated with several psychiatric comorbidities. A higher prevalence of substance use disorders, cluster B personality disorders, bipolar disorders, anxiety disorders, and suicidality has been evidenced in ED purging patients [30,31,32,33]. Interestingly, individuals suffering from EDs with purging behaviors showed higher levels of perfectionism [34,35,36,37]. High-perfectionist individuals suffering from EDs displayed higher levels of the “bulimia” dimension of the Eating Disorders Inventory-2 (EDI-2) [35,37]. Higher perfectionism was identified among adolescents with EDs with binge–purging symptoms [34]. A large meta-analysis concluded significantly higher levels of perfectionism in AN-BP compared to AN-R [36]. Further, purging behaviors have also displayed an association with PTSD [31,33,38]. In these patients, it is thought that purging behaviors might have a functional role in managing PTSD symptoms and reducing negative effects, particularly in avoiding hyperarousal [33]. A recent study ran a network analysis of inpatients, studying perfectionism, self-esteem, and affective symptoms in anorexia nervosa subtypes, where the bridge centrality of concern over mistakes and self-esteem suggests a link between perfectionism, mood, and ED symptoms. Yet, this research has divided inpatients based on diagnosis. Thus, a more detailed understanding could be borrowed from the study of the connection between general eating and eating psychopathology and the specific symptom presentation, such as the presence of purging symptoms [39].

Given the heavy medical consequences of purging, the interruption of drug abuse and vomiting is a priority goal in the path of care of EDs. First, since patient compliance is essential for improvement, adequate psychoeducational instructions on gastrointestinal functioning must be warranted. It must always be considered that the purging interruption time is particularly challenging for patients. As a rebound effect, they might report constipation and edema, usually lasting some weeks. Further, because of the physiological weight regain, patients may experience anxiety feelings. For these reasons, it is important to also provide psychological support and the strict monitoring of the actual abstinence of purging [40,41]. A treatment protocol for stopping purging has not been officially established yet. Among those promoted in past years, Harper et al. [42] proposed to substitute stimulant laxatives with a mixture of natural stool softeners and 30 mL of Magnolax, progressively reducing that by 5 mL every 3–7 days until suspension. Both in AN and BN, the core treatment for purging patients is considered Cognitive Behavioral Therapy (CBT) [43,44] or Enhanced CBT (E-CBT) [45,46]. Pharmacotherapy plays a secondary role in purging behaviors compared to psychotherapy. Complete abstinence from binging and purging when treated with antidepressants occurred in only 20 to 30% [43]. The most utilized antidepressants are fluoxetine and sertraline [47,48,49]. However, preliminary promising results have been obtained with naltrexone [50] and stimulants [51]. In recent years, new approaches to reduce purging have been proposed. Exposure and response prevention therapy (ERP) consists of exposing individuals to a binge and preventing purging behaviors so as to remove the threat of weight gain associated with binge eating. Several small single-case design studies found some reduction in purging utilizing this technique. However, it seemed to provide only a marginal benefit when compared to CBT [52]. Virtual reality (VR) enhanced CBT has been proven to be non-significant in decreasing the frequency of purges compared to CBT [53]. So far, CBT is the reference treatment option for ED purging patients [54,55].

When purging behaviors cause medical complications, or comorbidities precipitate, hospitalization is required [47,56,57]. Intensive inpatient treatment has been shown to significantly reduce the severity of binge eating and compensatory behavior, overall eating disorder symptom severity, and overall psychopathology, even when controlling for antidepressant medication [58,59,60,61]. The rationale of hospitalization is to provide a structured setting in which to eat adequate meals is useful to break the vicious circle of restrictions and binge purging episodes. At the same time, hospitalization should target the symptomatology of comorbidities. To do so, the planning of an eating scheme, periodic medical examinations to recover from complications, and an adjustment of pharmacotherapy are essential. Also, monitoring at meals, post meals, and during the day is required to ensure individuals interrupt their purging behaviors. However, a description of specific interventions in the context of hospitalization is still lacking [62]. For instance, as described above, psychotherapy has a central role in the cure of purging behaviors, but no evidence in the literature has assessed whether it is effective in a hospital context. Also, no studies have considered the most useful pharmacological options for acute symptomatology. So far, the clinical appearance and treatment of purging behaviors, specifically in the context of hospitalization, are important gaps that need to be filled by research.

### 1.2. Aims of the Study

The previous literature on this topic suggests an overall more complicated clinical picture for EDs with purging behavior. To confirm and assess this evidence, individuals suffering from ED were compared based on the purging behavior, both on a clinical and psychopathological level. The analyzed variables included anamnestic information, clinical data regarding the current hospitalization, and eating and general psychopathology. Since the previous literature highlighted an association between purging behaviors and perfectionism, this psychopathological dimension was also included. To our knowledge, all studies available in the literature focused on diagnostic subtypes, considering the different purging behaviors as a whole. So far, no evidence analyzed the possible differences between the specific types of purging. For this reason, participants were classified according to no purging behaviors, vomiting, the abuse of laxatives, and both vomiting and the abuse of laxatives. Since evidence on inpatients with purging behavior is scarce, the present study was conducted specifically in the context of hospitalization. Among the other information included, whether purging behaviors influenced clinical and psychological outcomes was evaluated. The focus of this research was expressly on purging symptoms rather than ED subtypes.

The aim of the study was to assess whether significant differences emerged among ED inpatients based on the specific type of purging behavior: no purging, vomiting, laxative abuse, or both vomiting and laxative abuse. Based on the previous literature, the groups of ED inpatients with purging behavior were expected to show a worse clinical picture and greater general and eating psychopathology with respect to the group without. So far, no evidence is available regarding the possible differences based on the specific type of purging behavior.

The purpose of the present research was to collect information on clinical data, eating psychopathology, general psychopathology, and perfectionism among individuals suffering from EDs based on the specific type of purging behavior separately; to assess whether differences between individuals with purging and no purging behaviors emerged; and to assess whether differences based on the specific type of purging behavior appeared. 

## 2. Materials and Methods

### 2.1. Participants

From 2015 to 2023, 302 inpatients subsequently admitted at the inpatient setting of the Eating Disorders Center of the “Città della Salute e della Scienza” hospital of the University of Turin, Italy, were involved in the study. All participants were diagnosed with EDs according to DSM-5 [2] by trained psychiatrists based on the Structured Clinical Interview for DSM-5 [63]. Inclusion criteria were set as follows: (a) aged between 18 and 65 years old; (b) had a confirmed diagnosis of ED according to the Structured Clinical Interview for the Diagnostic and Statistical Manual of Mental Disorders, 5th edition [63]; (c) had the capacity of reading the Italian language; and (d) had no current or past psychotic or bipolar disorder or current substance use disorder.

No individuals refused to fill in the questionnaires proposed. All participants voluntarily agreed to take part in the present study through written informed consent according to the Ethical Committee of our institution. The Ethical Committee approved the present retrospective study under registration number 00295/2021 of 9/6/2021.

### 2.2. Procedures and Measures

Upon admission, experienced nursing personnel measured participants in height and weight. Then, a trained psychiatrist interviewed individuals, collecting clinical and demographic data. Each participant provided information regarding age, gender, marital status, housing solution, ethnicity, years of illness, lowest weight reached, previous hospitalizations, previous pharmacological treatments, caloric intake, vomiting, laxative abuse, diuretic abuse, physical hyperactivity, self-harm, and previous suicidal attempts. The detection of purging symptoms in this study did not take into account the DSM-5 cut-offs for obtaining the diagnostic threshold. Instead, even the occasional presence of purging symptoms in the previous weeks was considered.

All participants were asked to complete the following self-report questionnaires, which are among the most utilized to assess eating and general psychopathology in EDs:Eating Disorder Examination Questionnaire (EDE-Q; [64]). This questionnaire was included to assess eating psychopathology during the previous 28 days. It consists of 28 items producing a total score and four subscales: “dietary restraint”, “eating concerns”, “weight concerns”, and “shape concerns”. The Italian validation of this questionnaire provided high internal consistency, with a Cronbach alpha value > 0.90 [65].Frost Multidimensional Perfectionism Scale (F-MPS; [66]). This tool was utilized to explore the main aspects of perfectionism. Moreover, 35 items give a total score and there are six subscales as follows: “concern over mistakes”, “personal standards”, “parental expectations”, “parental criticism”, “doubts about actions” and “organization” and a total score. Higher scores correspond to higher perfectionism traits. The Italian version of the questionnaire showed good internal consistency, with a Cronbach alpha value > 0.75 [67].State–Trait Anxiety Inventory (STAI; [68]). This questionnaire was employed to have a measure of anxiety levels. Moreover, 20 items assess anxiety in the present moment (“state anxiety”) and 20 items the basal levels of anxiety (“trait anxiety”). Higher scores correspond to higher levels of anxiety. The internal consistency of the Italian version of the tool is good, with Cronbach alpha values between 0.86 and 0.95 [69].Beck Depression Inventory (BDI; [70]). This tool assessed depression levels. Furthermore, 21 items produce a total score, where higher scores indicate greater depression with good internal consistency for the Italian version (Cronbach alpha = 0.87; test–retest reliability > 0.70; [71]).

All patients were asked to complete the questionnaires during the first days of hospitalization so that treatment did not influence them. Only for the EDE-Q were patients asked to fill out the questionnaire a second time in the days before discharge to derive the difference in its scores between hospital admission and discharge (ΔEDE-Q).

During hospitalization, inpatients receive a diet based on their physical condition. Trained psychiatrists interview individuals daily to offer support, assess their clinical progress, adjust pharmacological therapy, and ask about possible difficulties. At least once a week, blood tests and weight are monitored, and specialized nutritionists conduct a complete medical examination.

At the end of the hospitalization, nursing personnel measured individuals’ weight to derive the difference in BMI between hospital admission and discharge (ΔBMI), and the authors collected information on caloric intake at discharge. The length of the hospitalization and the drop-outs from the treatment protocol were reported.

### 2.3. Statistical Analysis

The statistical analyses were performed utilizing IBM SPSS 29.0 Statistics Software (SPSS).

ED individuals were divided into four groups based on free-from-purging behavior (NP), the presence of vomiting (V), the abuse of laxatives (L), and both vomiting and the abuse of laxatives (V + L). Since the interest of the study was in symptoms rather than diagnosis, the classification was based on the presence of the behavior regardless of its frequency. Purging behavior was considered positive if present in the last month, although sporadically. SPSS power analysis for ANOVA was used to estimate the sample size, with an alpha level of 5%. This analysis shows a power greater or higher than 80% when the sample size is above 112 participants.

A one-way ANOVA test was utilized to determine possible differences concerning the variables considered. All variables were continuous. The limit for statistical significance was set at *p * =  0.05, then an overconservative Bonferroni–Holm correction for multiple comparisons was applied, and thus, *p* = 0.0018. Subsequently, a Bonferroni post hoc analysis was performed in order to compare the four groups with each other.

## 3. Results

The clinical sample was composed of 302 individuals. Moreover, 169 individuals were diagnosed with the restricting AN (AN-R) subtype, 80 with the binge–purging subtype (AN-BP), and 53 with BN (see Appendix A). The characteristics of the sample are detailed in Table 1. Further information (i.e., gender distribution, ethnic composition, housing solutions, marital status, the type of access to the ED Unit, and purging symptom distribution among diagnoses) is available in the Appendix A.

### Comparison between ED Inpatients Based on Purging Behavior

Differences between ED inpatients with no purging behavior, vomiting, laxative abuse, and vomiting + laxative abuse are described in Table 2.

In total, 171 individuals had no purging behavior (NP), 82 reported vomiting (V), 20 reported laxative abuse (L), and 29 both vomiting and laxative abuse (V + L). 

After Bonferroni’s correction (the level of significance *p* = 0.0018), the ANOVA test evidenced significant differences in the four groups in age (F = 7.500, *p* < 0.001), years of illness (F = 10.729, *p* < 0.001), BMI at discharge (F = 14.602, *p* < 0.001), STAI state anxiety (F = 11.154, *p* < 0.001), STAI trait anxiety (F = 10.377, *p* < 0.001), BDI (F = 6.876, *p* < 0.001), EDE-Q eating concerns (F = 14.049, *p* < 0.001), EDE-Q shape concerns (F = 11.112, *p* < 0.001), EDE-Q weight concerns (F = 12.790, *p* < 0.001), EDE-Q global score (F = 12.737, *p* < 0.001), and F-MPS parental criticism (F = 9.569, *p* < 0.001). 

No differences emerged concerning the length of stay during current hospitalization (F = 4.328, *p* = 0.005), BMI at admission (F = 0.531, *p* = 0.661), caloric intake at admission (F = 0.436, *p* = 0.727), lowest weight (F = 2.730, *p* = 0.044), number of previous hospitalizations (F = 5.117, *p* = 0.002), caloric intake at discharge (F = 4.926, *p* = 0.002), EDE-Q dietary restraint (F = 4.600, *p* = 0.004), F-MPS concern over mistakes (F = 4.604, *p* = 0.004), F-MPS personal standards (F = 0.339, *p* = 0.797), F-MPS parental expectations (F = 2.436, *p* = 0.065), F-MPS doubts about actions (F = 3.858, *p* = 0.010), F-MPS organization (F = 2.823, *p* = 0.039), F-MPS global score (F = 3.895, *p* = 0.009), ΔBMI (F = 1.309, *p* = 0.275), ΔEDE-Q global score (F = 0.781, *p* = 0.507), and drop-outs during current hospitalization (F = 2.401, *p* = 0.067).

The Bonferroni post hoc analysis provided information upon comparing the groups with each other. 

NP and V were younger than L. NP had fewer years of illness compared to L and V + L. L had more years of illness compared to V. NP displayed a lower BMI at discharge compared to V and V + L.

NP displayed overall significantly lower scores on eating and general psychopathology compared to the other groups. Specifically, the STAI state and trait anxiety scores were lower compared to V and L; the BDI score was lower compared to V; the EDE-Q eating concern and shape concern scores were lower than V and V + L; the EDE-Q weight concern and global scores were lower than groups V, L, and V + L; and the F-MPS parental criticism score was lower than V and V + L.

## 4. Discussion

This study aimed to provide an investigation of purging behaviors in a sample of inpatients in a specialist ward for the treatment of eating disorders. All studies available in the literature considered the different types of purging behavior as a whole. So far, no evidence analyzed the possible differences between the specific types of purging. Unlike previous research, the interest of the present research was analyzing the symptoms and the different types of purging behaviors rather than the ED subdiagnosis. For this reason, persons suffering from EDs were classified in no purging behaviors (NP), vomiting (V), the abuse of laxatives (L), and both vomiting and the abuse of laxatives (V + L).

The three groups of ED inpatients with purging behaviors vs. inpatients without any purging behaviors displayed overall a more complicated clinical picture and worse psychopathology, but no differences emerged between the different types of purging behavior. These findings suggest that purging behavior is a marker of severity in EDs independently of the specific purging behavior. The different behaviors might have similar consequences on individuals or derive from common psychopathological pathways. In this regard, all the specific purging symptoms might be due to common motivations, like the drive for thinness, self-punishment, or emotion regulation [23]. Also, some psychopathological characteristics might incline persons to purge in general, and the specific behavior will depend on idiosyncratic events or individual preferences. The comparable severity between purging behaviors recommends that prevention programs seek all purging symptoms and not to underestimate, even when a single purging symptom is present (e.g., laxative abuse).

Those in the NP group were younger than those in the L group and had fewer years of illness compared to L and V + L. As the literature has highlighted the common progression of AN-R into purging disorders (AN-BP and BN) [3,4], it could be surmised that the disease affecting those of a younger age and a longer duration of the disease might be consequences. In this regard, a recent large systematic review and meta-analysis [4] concluded that 41.84% of AN-R individuals undergo the transition to purging disorders at some point.

NP displayed overall significantly lower scores on eating and general psychopathology. These findings may reinforce the evidence of a higher psychopathological severity of AN-BP compared to AN-R individuals [3,72,73]. Purging behaviors per se might also produce a sense of shame in individuals, which could in turn be responsible for higher levels of depression and anxiety [74]. Purging behavior is associated with worse psychopathology in other EDs as well. BN individuals displayed worse psychopathological scores compared to binge eating disorder (BED) individuals [75,76,77]. Also, a higher prevalence of comorbidities has been evidenced in ED purging patients [30,31,32,34,38,78]. In previous research, individuals with AN-BP exhibited significantly lower attention/vigilance scores compared to those with AN-R, suggesting that purging behavior could affect even neurocognitive impairment [79]. 

NP showed lower levels of the “parental criticism” dimension of perfectionism when confronted with V and V + L. This finding is also aligned with previous evidence on this topic. A recent study [34] on 178 adolescents with EDs showed higher perfectionism among those with binge–purging symptoms. Longo et al. [35,37] divided individuals with EDs based on overall perfectionism. In both studies, the low-perfectionism group displayed lower levels of the “bulimia” dimension of EDI-2 compared to the high-perfectionism group. Finally, a large meta-analysis [36] evidenced in AN-BP significantly higher levels of perfectionism compared to AN-R. Some authors hypothesized that binging episodes might be perceived as a failure, with consequent bad feelings producing an urge to regain control through purging behaviors [34,80]. Additionally, to this view, it might also be surmised that purging behaviors, in some cases, act as a dysfunctional way to manage the emotional stress deriving from marked perfectionism. Higher “parental criticism” scores might also reflect the internalization of parental critical attitudes during early childhood interactions. Since early life experiences like emotional abuse act as a risk factor for the purging phenotype of eating disorders, this might at least partially explain the finding [81]. Also, parental criticism was linked to lower parental educational levels [82].

No significant differences emerged between the groups in the other dimensions and the global value of perfectionism. Possibly, the small numerousness of the sample prevented us from evidencing significant differences. Further research with enlarged samples of individuals with EDs is required to verify whether significant differences also appear in the other dimensions of perfectionism.

No difference based on purging behaviors emerged concerning ΔBMI and ΔEDE-Q global score. These findings are encouraging. If inpatients with purging behaviors, when confronted with the groups of non-purging inpatients, are overall more critical from several points of view, hospitalization seems to be equally efficient. This indicates that the protocols of treatment during hospitalization are solid enough to compensate for the difficulties of individuals with purging behaviors. Often clinicians propose to individuals with decades-long EDs with multiple comorbidities (severe enduring anorexia nervosa—SEAN) low-intensity treatment approaches, pushed by the low odds of effective cures [83]. Based on the research findings, this should not be applied to individuals with purging behavior, even if they show a worse clinical state. However, the results might also be due to the short measurement period (the mean length of stay during current hospitalization: 33.8 days), and possible differences could arise during a follow-up. Both BMI and eating psychopathology require prolonged time to improve. Future studies with proper follow-up after hospitalization may solve this discussion. 

Ultimately, the findings of the present study give a picture of ED inpatients with purging behaviors characterized by a longer duration of illness, worse clinical state, and worse psychopathology. A large retrospective study [84] evidenced that longer illness duration, a higher maximum BMI, higher novelty seeking, and lower self-directedness were significantly associated with the crossover from AN-R to BN. Also, patients with BN with initial AN-R exhibited a lower desired body weight and higher drive for thinness, asceticism, and social insecurity scores compared to patients with BN with no history of AN-R, advancing that they could constitute two clinically distinct subgroups with prognostic implications [84]. The addition of subthreshold purging behaviors for a person suffering from AN-R must be regarded as a considerable red flag. Most of the cases of ED onsets do not directly access the specialist ED unit but rather mental health professionals and general practitioners [85]. So, it is important to spread the notion that purging behaviors are signals of a severe clinical and psychopathological picture requiring specialized assessment [86]. From the perspective of preventing chronic development and early intervention in EDs, more attention may be allocated to cases presenting with purging behaviors. 

The aim of the present study was to investigate a sample of inpatients, grouping them using symptoms and not diagnosis and thus borrow new data on profiles of inpatients with ED. For mental health professionals specialized in EDs, the appearance of any purging behaviors (vomiting, laxative abuse, or both) during the follow-up of a person suffering from AN-R may suggest an intensification of the cure. Based on the findings of a worse clinical picture and psychopathology, it might be important for clinicians to intensify the treatment when purging behaviors appear. This could, for instance, imply shortening the time between medical examinations, calling for a nutritional consultation, reviewing the pharmacological treatment, considering a period of hospitalization, or enlarging the therapeutic options. Although the sample size was more than twice as large as required by the power analysis, since the overconservative Bonferroni–Holm correction was used to minimize the type 2 error, it is possible that the differences between the groups were underestimated; and thus, future studies may offer more indications with regard to differentiated and tailor-made hospital treatment for inpatients with different purging symptoms (e.g., laxative abuse and not vomiting).

Ultimately, in an increasingly tailored picture of the paths of care, together with the other socio-economical, clinical, and psychological features, purging behavior is a factor that should be properly considered in the definition of the best and most multidisciplinary approach to persons suffering from EDs. In fact, encounters with individuals in acute periods of crisis, such as those requiring hospitalization, may influence physicians’ emotional attitudes and thus clinical decisions. Specialists facing the challenges posed by individuals with severe ED, with typical resistance to treatment, must constantly monitor their own emotional states. Greater knowledge of clinical presentations can help therapists and teams in the delicate decision-making processes involved in treating these conditions [87].

## 5. Conclusions

In conclusion, the present study has the value of having assessed the difference between individuals suffering from EDs from the perspective of the specific type of purging behavior. In the context of increasingly more tailored paths of diagnosis and cures, the present research contributes to surpassing the model of fixed ED diagnoses currently questioned by some authors [4,5,6]. This approach allowed us to explore more precisely the specific impact of each purging behavior on clinical and psychopathological outcomes, providing a more detailed understanding of the pathological dynamics associated with purging behaviors. This paradigm opens the possibility for more tailored clinical interventions and might help to overcome some limitations of classical diagnostic categorization, which may not fully capture the complexity of disordered eating behaviors in persons with EDs. Also, the approach used in the present study could be applied to a wide range of individuals suffering from EDs, including those who do not fit neatly into the traditional diagnostic labels.

Some limitations of the present study must be acknowledged. First, the small size of the sample of inpatients abusing laxatives and both vomiting and abusing laxatives raises some doubts regarding the findings due to real invariance between the types of behavior or rather to the limited numerousness of the analysis. More research with enlarged samples is required to clear this discussion. Also, purging behaviors comprise diuretic abuse, which could not be considered due to an insufficient number of individuals with this problem in our center. Given the important medical consequences of diuretic abuse, it deserves proper investigation in future studies. Second, the purging behavior and the measures of psychological variables were self-reported. Particularly, some participants could have omitted information on purging behavior due to concerns about changes in their therapeutic course. Third, the cross-sectional study design did not allow us to conclude any causal relation but only statistical associations between purging behaviors and the variables considered. Fourthly, a power analysis and the Bonferroni–Holm correction were used, i.e., two tools that aim to minimize the type 2 error. However, they may increase the risk of committing a type 1 error, i.e., underestimating the differences between the two groups. Future studies can analyze, with an increased sample size, the differences between individuals with ED and different purging symptoms, which may have been underestimated in the present study. Also, individuals suffering from ED were considered in the context of hospitalization, hence constituting the most severe cases. Evidence might not be expandable to less severe presentations. Finally, the findings are based only on measurable outcomes. Qualitative research could provide further contributions, particularly more refined and indefinite outcomes, such as: insights into the vulnerability of the therapists, trust between patients, their families and psychiatrists, and therapy involvement [88].

To conclude, the present study has the value of filling a gap in the existing literature, assessing individuals in a naturalistic setting (i.e., hospitalization) where evidence is still scarce, and surpassing the focus on diagnostic differentiation. In the sample, ED inpatients with purging behaviors were characterized by a longer duration of illness, higher parental criticism, and worse depressive, anxious, and eating psychopathology. No differences emerged between the specific types of purging behavior, neither on a clinical nor a psychopathological level. Importantly, to our knowledge, the present research is the first in the literature to consider the specific types of purging behavior instead of diagnostic subtypes in inpatients suffering from EDs. It is desirable that the present study might inspire further research in this area, advancing our understanding of the underlying mechanisms of EDs.

## Figures and Tables

**Table 1 healthcare-12-01858-t001:** Characteristics of the ED sample.

ED Inpatients	Mean (SD)	Yes (%) N (%)	No N (%)
Vomiting	-	111 (36.8)	191 (63.2)
Laxative abuse	-	49 (16.2)	253 (83.8)
Diuretic abuse	-	15 (5.0)	287 (95.0)
Age (years)	24.7 (8.7)	-	-
Length of stay during current hospitalization (days)	33.8 (19.2)	-	-
Years of illness (years)	7.1 (7.8)	-	-
BMI at admission (kg/m^2^)	15.34 (3.1)	-	-
Caloric intake at admission (kcal/day)	770.3 (451.4)	-	-
Lowest weight (kg)	36.3 (6.2)	-	-
Number of previous hospitalizations	2.4 (3.7)	-	-
BMI at discharge (kg/m^2^)	15.8 (2.8)	-	-
Caloric intake at discharge (kcal/day)	1482.5 (363.3)	-	-
ΔBMI (kg/m^2^)	0.5 (1.2)	-	-
ΔEDE-Q, global score	−0.6 (1.0)	-	-
Physical hyperactivity	-	142 (47.0)	160 (53.0)
Self-harm	-	58 (19.2)	244 (80.8)
Suicidal attempts	-	48 (15.9)	254 (84.1)
Previous failure of psychopharmacotherapies	-	139 (46)	163 (54)
Drop out from actual inpatient treatment	-	8 (2.6)	294 (97.4)

Legend: EDs = eating disorders; SD = standard deviation; N = numerousness; BMI = body mass index; ΔBMI = variation in BMI during the current hospitalization; EDE-Q = Eating Disorder Examination Questionnaire; ΔEDE-Q = variation in the EDE-Q global score during the current hospitalization.

**Table 2 healthcare-12-01858-t002:** Comparison between ED inpatients with no purging behavior, vomiting, laxative abuse, and both vomiting and laxative abuse.

ED Inpatients	No Purging (NP) (N = 171) Mean (SD)	Vomiting (V) (N = 82) Mean (SD)	Laxatives (L) (N = 20) Mean (SD)	Vomiting + Laxatives (V + L) (N = 29) Mean (SD)	F	*p*	Post Hoc Analysis
**Age**	**23.6 (8.3)**	**24.2 (8.1)**	**32.7 (12.2)**	**27.1 (7.3)**	**7.500**	**<0.001**	**(NP), (V) < (L)**
Length of stay during current hospitalization (in days)	37.1 (20.5)	30.1 (16.9)	31.3 (17.4)	26.5 (13.7)	4.328	0.005	
**Years of illness**	**5.5 (6.6)**	**7.8 (6.9)**	**14.4 (13.6)**	**10.0 (7.9)**	**10.729**	**<0.001**	**(NP) < (L), (V + L)** **(NP), (V) < (L)**
BMI at admission	15.2 (12.8)	16.9 (3.7)	15.5 (3.6)	16.3 (3.6)	0.531	0.661	-
Caloric intake at admission (kcal/day)	773.7 (420.5)	767.0 (492.8)	851.7 (659.7)	688.6 (345.7)	0.436	0.727	-
Lowest weight	35.6 (5.6)	37.6 (6.8)	35.0 (6.2)	37.8 (7.0)	2.730	0.044	-
Number of previous hospitalizations in anamnesis	1.7 (3.1)	3.0 (3.2)	4.0 (5.6)	3.5 (5.4)	5.117	0.002	-
**BMI at discharge**	**15.0 (2.1)**	**17.3 (3.2)**	**15.8 (2.4)**	**16.6 (3.6)**	**14.602**	**<0.001**	**(NP) < (V), (V + L)**
Caloric intake at discharge (kcal/day)	1548.2 (342.4)	1386.1 (382.0)	1493.3 (385.1)	1362.1 (342.7)	4.926	0.002	
**STAI State Anxiety**	**53.6 (13.9)**	**63.4 (11.3)**	**62.8 (13.2)**	**55.6 (13.8)**	**11.154**	**<0.001**	**(NP) < (V), (L) (NP), (V + L) < (V)**
**STAI Trait Anxiety**	**55.9 (13.9)**	**65.6 (12.2)**	**64.8 (12.6)**	**59.9 (14.9)**	**10.377**	**<0.001**	**(NP) < (V), (L)**
**BDI, total score**	**15.7 (8.4)**	**20.6 (8.1)**	**20.1 (8.2)**	**17.8 (7.5)**	**6.876**	**<0.001**	**(NP) < (V)**
EDE-Q, dietary restraint	2.9 (2.0)	3.6 (1.9)	3.7 (2.2)	4.2 (1.6)	4.600	0.004	
**EDE-Q, eating concerns**	**2.6 (1.5)**	**4.0 (2.3)**	**3.6 (1.5)**	**3.8 (1.2)**	**14.049**	**<0.001**	**(NP) < (V), (V + L)**
**EDE-Q, shape concerns**	**3.8 (1.7)**	**5.0 (1.8)**	**4.8 (1.6)**	**5.0 (1.3)**	**11.112**	**<0.001**	**(NP) < (V), (V + L)**
**EDE-Q, weight concerns**	**3.2 (1.8)**	**4.5 (1.9)**	**4.5 (1.6)**	**4.5 (1.5)**	**12.790**	**<0.001**	**(NP) < (V), (L), (V + L)**
**EDE-Q, global score**	**3.1 (1.6)**	**4.2 (1.5)**	**4.2 (1.6)**	**4.4 (1.3)**	**12.737**	**<0.001**	**(NP) < (V), (L), (V + L)**
F-MPS, concern over mistakes	28.7 (10.7)	33.4 (9.6)	33.0 (11.3)	32.1 (9.7)	4.604	0.004	-
F-MPS, personal standards	25.1 (7.9)	24.2 (6.4)	24.6 (8.6)	25.5 (6.7)	0.339	0.797	-
F-MPS, parental expectations	10.2 (6.4)	11.3 (5.7)	13.5 (4.9)	12.3 (6.4)	2.436	0.065	-
**F-MPS, parental criticism**	**8.6 (3.9)**	**12.0 (7.2)**	**10.7 (3.5)**	**11.4 (4.1)**	**9.569**	**<0.001**	**(NP) < (V), (V + L)**
F-MPS, doubts about actions	12.6 (6.1)	14.5 (4.1)	13.0 (3.9)	16.1 (9.9)	3.858	0.010	-
F-MPS, organization	23.7 (5.2)	22.0 (5.8)	24.7 (5.7)	24.5 (5.5)	2.823	0.039	-
F-MPS, global score	108.6 (27.2)	117.3 (23.8)	121.2 (31.4)	122.0 (29.0)	3.895	0.009	-
ΔBMI	0.7 (0.8)	0.2 (1.2)	0.6 (2.6)	0.4 (0.8)	1.309	0.275	-
ΔEDE-Q, global score	−0.6 (0.9)	−0.4 (1.2)	−0.8 (1.0)	−0.9 (1.3)	0.781	0.507	-
Drop-outs during current hospitalization	0.004 (0.065)	0.028 (0.166)	0.000 (0.000)	0.053 (0.227)	2.401	0.067	

Legend: EDs = eating disorders; SD = standard deviation; F = F-value; *p* = *p*-value; BMI = body mass index; ΔBMI = variation in BMI during the current hospitalization; STAI = State–Trait Anxiety Inventory; BDI = Beck Depression Inventory; EDE-Q = Eating Disorder Examination Questionnaire; F-MPS = Frost Multidimensional Perfectionism Scale; ΔEDE-Q = variation in the EDE-Q global score during the current hospitalization; *p* = <0.0018 after Bonferroni’s correction; in bold when *p* < 0.0018.

## Data Availability

The corresponding author had full access to all the data in the study and takes responsibility for the integrity of the data and the accuracy of data analysis. Data sharing is not applicable since consent to publish was for aggregated data only. Code availability: atandard codes were used in the analysis. Sample codes will be shared upon demand.

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
