# Peer review of "The Clinical and Psychopathological Profile of Inpatients with Eating Disorders: Comparing Vomiting, Laxative Abuse, and Combined Purging Behaviors"

_healthcare, 2024, doi:10.3390/healthcare12181858_

Round 1

Reviewer 1 Report

Comments and Suggestions for Authors

The study provides an interesting characterization of patients with eating disorders and different purging behaviors, namely laxative abuse, vomiting, or both, compared to those without purging. The study is well-conducted, the methodology is appropriate, and the limitations are well-discussed. Some minor issues should be addressed by the authors for publication in this journal:

Introduction:

-please, more clearly provide the rationale for comparing the different groups in terms of levels of perfectionism.

Methods:

-I would suggest adding a power calculation

Discussion:

-I recommend discussing the negative results in light of a power calculation;

- Further exploration of the "parental criticism" dimension within the F-MPS questionnaire is suggested. This dimension appears to relate to the internalization of parental critical attitudes during early childhood interactions. Thus, this result should be discussed by considering not only the literature on perfectionism but also the role of early life experiences (particularly emotional abuse, of which parental criticism is one dimension) as a risk factor for the purging phenotype of eating disorders (for a review, see Rossi et al. 2024, The maltreated eco-phenotype of eating disorders: a new diagnostic specifier?)

- The following sentence is unclear: "NP showed lower levels of the 'parental criticism' dimension of perfectionism when compared with V and V+L. This result might be at least partially due to the psychological consequences of purging, like lower self-esteem and feelings of guilt." From what is reported, it seems that the dimension of parental criticism could be a consequence of purging, which is difficult to understand since the questionnaire items that make up this subscale refer to the internalization of parental attitudes dating back to the patient's childhood. I suggest to clarify the meaning of this sentence or remove it.

Author Response

The study provides an interesting characterization of patients with eating disorders and different purging behaviors, namely laxative abuse, vomiting, or both, compared to those without purging. The study is well-conducted, the methodology is appropriate, and the limitations are well-discussed. Some minor issues should be addressed by the authors for publication in this journal:

Introduction:

  • Please, more clearly provide the rationale for comparing the different groups in terms of levels of perfectionism.

Thank you for the comment. In the Introduction, we expanded the description of the previous literature on this topic, and we provided a rationale for the inclusion of this information in the analysis.

Methods:

  • I would suggest adding a power calculation

Thank you for the comment. Power calculation was performed and added.

Discussion:

  • I recommend discussing the negative results in light of a power calculation;

Thank you for the comment. The results of the power calculation have been discussed and a comment was added in the limitation section.

  • Further exploration of the "parental criticism" dimension within the F-MPS questionnaire is suggested. This dimension appears to relate to the internalization of parental critical attitudes during early childhood interactions. Thus, this result should be discussed by considering not only the literature on perfectionism but also the role of early life experiences (particularly emotional abuse, of which parental criticism is one dimension) as a risk factor for the purging phenotype of eating disorders (for a review, see Rossi et al. 2024, The maltreated eco-phenotype of eating disorders: a new diagnostic specifier?)

Thank you for the comment. This finding was further discussed as suggested.

  • The following sentence is unclear: "NP showed lower levels of the 'parental criticism' dimension of perfectionism when compared with V and V+L. This result might be at least partially due to the psychological consequences of purging, like lower self-esteem and feelings of guilt." From what is reported, it seems that the dimension of parental criticism could be a consequence of purging, which is difficult to understand since the questionnaire items that make up this subscale refer to the internalization of parental attitudes dating back to the patient's childhood. I suggest to clarify the meaning of this sentence or remove it.

Thank you for the comment. We realized the sentence was unclear, so we decided to remove it.

Reviewer 2 Report

Comments and Suggestions for Authors

Although this is an article with interesting findings, there are some points that need to be revised:

Basic statistical findiings with numbers are missing in the Abstract. Please add.

The hypothesis should be based on relevant literature. Please add.

Authors can add in the Introduction recent findings coming not only from quantitative, but also from qualitative research as these articles provide a more thorough approach and highlight the limitations of quantitative and how therapists understand and respond to the symptomatology (e.g. for a recent article on this https://doi.org/10.1080/13642537.2023.2278088). These points raised should be added in the Limitations section.

Why were these tests/questionnaires and variables included instead of others? Please justify.

Statistics must be presented in APA style in the main text as Tables are huge to grasp.

Please provide a more extended discussion and discuss the perspective of mental health experts treating these symptoms as this point could influence the perception/expression of emotions and thoughts as well as behaviors in the patients.

Comments on the Quality of English Language

Moderate English language editing.

Author Response

Although this is an article with interesting findings, there are some points that need to be revised:

  • Basic statistical findiings with numbers are missing in the Abstract. Please add.

Thank you for the comment. We added this information to the Abstract.

  • The hypothesis should be based on relevant literature. Please add.

Thank you for the comment. The Introduction has been enriched and refined, also adding data from Delaquis et al., 2024. Based also on the comment of Reviewer 1, the rationale behind the inclusion of perfectionism in the analysis has been clarified.

  • Authors can add in the Introduction recent findings coming not only from quantitative, but also from qualitative research as these articles provide a more thorough approach and highlight the limitations of quantitative and how therapists understand and respond to the symptomatology (e.g. for a recent article on this https://doi.org/10.1080/13642537.2023.2278088). These points raised should be added in the Limitations section.

Thank you for the comment. The Introduction has been enriched, and this point has been added to the Limitations.

  • Why were these tests/questionnaires and variables included instead of others? Please justify.

Thank you for the comment. We decided to include the most utilized tools to assess eating and general psychopathology in the context of EDs. We added a sentence to the text to underline this motivation. Further, based on the comment of Revisor 1, we expanded the reasons to include perfectionism in the analysis.

  • Statistics must be presented in APA style in the main text as Tables are huge to grasp.

Thank you for the comment. F-values and p-values have been added.

  • Please provide a more extended discussion and discuss the perspective of mental health experts treating these symptoms as this point could influence the perception/expression of emotions and thoughts as well as behaviors in the patients.

Thank you for the comment. The Discussion has been enriched in the light of this suggestion.

Reviewer 3 Report

Comments and Suggestions for Authors

In this paper, the authors used the ANOVA to compare the psychopathological features of four ED groups: no purging, vomiting, laxative abuse, and both vomiting and laxative abuse. The authors found higher levels of psychopathology in groups with purging behaviors, and found no differences between the groups with different types of purging behaviors (vomiting or laxative abuse). This paper is well-written and fits well with Healthcare. However, I am slightly concerned about how well the statistical methods can link to the interpretations and conclusions. 

My main concern is that the authors tried to argue no differences between groups with different types of purging behaviors (vomiting and laxative abuse). This argument is based on the fact that the corresponding statistical tests had p-values above the defined cut-off or alpha level (p=.0018). First, although the p-value below a cut-off indicates the acceptance of the alternative hypothesis (there are true differences), the p-value above a cut-off does not indicate the acceptance of the null hypothesis (there are no differences). In addition, the authors used the over conservative Bonferroni-Holm correction for multiple comparisons to determine the cut-of as p=.0018. This makes it even harder to reject the null and accept the alternative hypothesis. For example, in Table 2, quite a few tests have p-values between .0018 and .05 but were not further explored due to p>.0018. Although the Bonferroni-Holm correction, being highly conservative, can reduce the Type I errors, it also simultaneously boost the Type II errors by failing to detect true differences. As the authors were trying to argue no differences between groups, the increased Type II error rate is a problem for this study. 

Other questions:

1. How did the authors define the four purging groups (NP, V, L, V+L)? Is it simply by the presence of symptoms regardless of frequency?

2. What are the distributions of purging groups (NP, V, L, V+L) within each ED diagnosis (AN-R, AN-BP, BN)? The authors found psychopathological differences between purging and non-purging groups. Could these differences be due to non-purging patients having a less severe ED instead of differences in purging behaviors?

3. Table 1: The mean BMI at discharge is 15.8, which seems abnormally low for discharged patients.

4. Is it possible for authors to provide more detailed information of the timing of different measurements? As the patients are receiving treatment in hospital settings, I anticipate that individuals with more severe conditions are likely to receive more intense care. As a result, if the assessments were performed during treatment or at discharge, I think that they would be confounded by the treatment effects.

Author Response

In this paper, the authors used the ANOVA to compare the psychopathological features of four ED groups: no purging, vomiting, laxative abuse, and both vomiting and laxative abuse. The authors found higher levels of psychopathology in groups with purging behaviors, and found no differences between the groups with different types of purging behaviors (vomiting or laxative abuse). This paper is well-written and fits well with Healthcare. However, I am slightly concerned about how well the statistical methods can link to the interpretations and conclusions. 

  • My main concern is that the authors tried to argue no differences between groups with different types of purging behaviors (vomiting and laxative abuse). This argument is based on the fact that the corresponding statistical tests had p-values above the defined cut-off or alpha level (p=.0018). First, although the p-value below a cut-off indicates the acceptance of the alternative hypothesis (there are true differences), the p-value above a cut-off does not indicate the acceptance of the null hypothesis (there are no differences). In addition, the authors used the over conservative Bonferroni-Holm correction for multiple comparisons to determine the cut-of as p=.0018. This makes it even harder to reject the null and accept the alternative hypothesis. For example, in Table 2, quite a few tests have p-values between .0018 and .05 but were not further explored due to p>.0018. Although the Bonferroni-Holm correction, being highly conservative, can reduce the Type I errors, it also simultaneously boost the Type II errors by failing to detect true differences. As the authors were trying to argue no differences between groups, the increased Type II error rate is a problem for this study.

Thank you for the comment. These considerations have been discussed and commented in the limitation section.

Other questions:

  • How did the authors define the four purging groups (NP, V, L, V+L)? Is it simply by the presence of symptoms regardless of frequency?

Thank you for the comment. We were interested in symptoms rather than diagnosis. Consequently, purging behavior was considered positive if present in the last month regardless of its frequency. We specified this aspect in the paragraph: “2.3 Statistical analysis”. 

  • What are the distributions of purging groups (NP, V, L, V+L) within each ED diagnosis (AN-R, AN-BP, BN)? The authors found psychopathological differences between purging and non-purging groups. Could these differences be due to non-purging patients having a less severe ED instead of differences in purging behaviors?

Thank you for the comment. The distribution of purging groups within ED diagnosis is detailed in Supplementary Material. As recognized among the limits, our study is exploratory. We identified a worse clinical and psychopathological picture among individuals with purging behaviors. Yet, the study design did not allow us to conclude any causal relation but only statistical associations between variables. We have also stressed in the paper that a population of inpatients is not representative of a general population of individuals with ED but of a smaller category of individuals requiring intensive care.

  • Table 1: The mean BMI at discharge is 15.8, which seems abnormally low for discharged patients.

Thank you for the comment. Our Eating Disorder Center at the University of Turin takes care of the most severe cases of Anorexia Nervosa. The less severe ones are followed by general psychiatrists. Consequently, the mean BMI of our patients is low. This data was reported also in other studies of our research group (10.3390/nu16081156, 10.1002/cpp.2931, 10.3390/jcm11226683).

In the Conclusions, we recognized a limit. The evidence that emerged in our analysis might not be expandable to less severe presentations. Furthermore, hospitalization is useful for patients in the acute phase. Discharge from our hospital often involves the transition to residential rehabilitation care or semi-residential intensive care, such as DH.

  • Is it possible for authors to provide more detailed information of the timing of different measurements? As the patients are receiving treatment in hospital settings, I anticipate that individuals with more severe conditions are likely to receive more intense care. As a result, if the assessments were performed during treatment or at discharge, I think that they would be confounded by the treatment effects.

Thank you for the comment. Details on the timing of questionnaire compilation have been added to the paragraph: “2.2 Procedures and measures”. All patients were asked to complete the questionnaires during the first days of hospitalization so that treatment did not influence them. Only for the EDE-Q, it was asked to fill out a second time in the days before discharge, to derive the difference in its scores between hospital admission and discharge (DEDE-Q).